# Identification of Animal-Based Welfare Indicators in Captive Reptiles: A Delphi Consultation Survey

**DOI:** 10.3390/ani11072010

**Published:** 2021-07-05

**Authors:** Alexandra L. Whittaker, Brianna Golder-Dewar, Jordyn L. Triggs, Sally L. Sherwen, David J. McLelland

**Affiliations:** 1Roseworthy Campus, School of Animal and Veterinary Sciences, The University of Adelaide, Roseworthy, SA 5371, Australia; brianna.golder-dewar@student.adelaide.edu.au (B.G.-D.); jordyn.triggs@student.adelaide.edu.au (J.L.T.); dmclelland@zoossa.com.au (D.J.M.); 2Wildlife Conservation and Science, Zoos Victoria, Melbourne, VIC 3052, Australia; ssherwen@zoo.org.au; 3The Animal Welfare Science Centre, The University of Melbourne, Melbourne, VIC 3052, Australia; 4Zoos South Australia, Frome Road, Adelaide, SA 5000, Australia

**Keywords:** reptile, animal-based assessment, Delphi, welfare

## Abstract

**Simple Summary:**

Reptiles are commonly housed in human care in zoo and wildlife parks, and as pets. In comparison to many other taxa, especially mammals, there is less known about their behavioral repertoire and how this may be used to assess their welfare. Furthermore, there is a current focus of zoos and wildlife parks to instigate formal assessment of welfare using species-appropriate welfare assessment tools. These tools should ideally comprise a mixture of resource-based and animal-focused indicators of welfare. Since there has been little consideration of animal-based indicators of welfare in reptiles, this study utilized a method of expert consultation (Delphi method) to gather opinions as to the validity and reliability of a range of animal-based criteria for assessment of reptile welfare. The resulting list of criteria comprises both health indicators and behavior-based measures. Further validation of these indicators in practical scenarios is now required to develop reptile-specific welfare assessment tools for use in zoos.

**Abstract:**

There is an increasing focus on evidence-based welfare assessment by animal care staff in zoos, along with a strong interest in animal welfare by the zoo-visiting public, to the extent that this can influence their choice of institutions to visit. Regulatory oversight of animal welfare standards continues to strengthen across many jurisdictions. Zoos are increasingly formalizing their practices with the development and refinement of evidence-based welfare assessment tools. There has been a drive for welfare assessment tools to comprise both resource-based and animal-based measures. However, animal-based indicators are not always well characterized, in terms of their nature and whether they infer a positive or negative affective state. This is especially so for reptiles, which are often considered behaviorally inexpressive and are under-researched. In this study, a Delphi consultation approach was used to gather expert opinion on the suitability of potential animal-based indicators of welfare for inclusion in a welfare assessment tool across four families of reptiles: Agamidae, Chelidae, Pythonidae, and Testudinidae. Two rounds of online surveys were conducted eliciting responses from a global group of professionals who work with reptiles. In the first survey, respondents were provided with an author-derived list of potential animal-based indicators for consideration of their validity and practicality as welfare indicators. The indicators were refined for the second survey including only those indicators that were considered valid or practical on the first survey (≥4 on a 5-point Likert scale), and that achieved ≥70% consensus amongst experts. In the second survey, respondents were asked to re-evaluate the reliability and practicality of the indicators and to rank them on these facets. Eight to ten assessment indicators for each family of reptiles were identified from Survey 2. These indicators were often health related, for example, presence of oculo-nasal discharge or wounds. However, some true behavioral indicators were identified, such as showing species-specific interest and alertness. These indicators should now be incorporated into taxon-tailored welfare assessment tools for trial and validation in captive reptile populations. This study provides a next step towards developing reptile-specific animal welfare assessment tools for these often-overlooked animals.

## 1. Introduction

Good zoos in all parts of the world are committed to providing high standards of welfare for the animals under their care. There is an increasing focus on evidence-based welfare assessment by animal care staff in zoos. At the same time, there is now strong public interest in animal welfare across all areas of animal use, with the expectation that high standards of animal welfare are sought and achieved [1,2]. Zoos and wildlife parks are perhaps held more accountable than other industries since they are accessed by the public, who then make determinations, rightly or wrongly, on animal welfare status. As a result, there has been considerable focus in recent years on the development and validation of methods to assess the welfare of zoo animals. Furthermore, formal assessment of welfare status has become an increasingly important component of zoo accreditation schemes, such as that of the Association of Zoos and Aquariums [3], and the Australasian Zoo and Aquarium Association [4]. There has also been a shift within the zoo community to consider supplementation of the traditional resource-based approach, to institutional welfare assessments, with the use of animal-based measures [5]. Reptiles are also commonly kept as pets. The ability to objectively assess reptile welfare will be of interest to many pet owners committed to the well-being of their pet, as well as jurisdictions regulating responsible pet ownership.

A recently proposed definition of animal welfare that incorporates current scientific knowledge based on a multidisciplinary approach is that:

‘the welfare of an animal is its positive mental and physical state as related to the fulfilment of its physiological and behavioural needs in addition to its expectations. This state can vary depending on the animal’s perception of a given situation.’ [6]

The term animal welfare is generally taken to refer to a long-lasting state made up of the summed experiences of an animal [7]. These experiences are frequently defined in terms of affective states. Affective states describe emotions such as joy, fear and happiness [8], and are usually characterized based on two dimensions: that of valence (direction, e.g., positive or negative), and strength or arousal [8,9]. Welfare determinations are made taking into consideration the relative number of so-called ‘positive’ states occurring, compared to the number of ‘negative’ states, with good welfare being defined when there are relatively more positive events than negative events over a given period, poor welfare when the opposite occurs, and a neutral state when positive and negative events are experienced equally [10].

Animal welfare can be assessed through a variety of means, such as examination of physiological, immunological, and behavioral coping responses to an event or environment [11]. However, at the current time, these measures are relatively more extensive, and developed, for the assessment of negative emotions such as pain or fear than they are for positive emotions. This presents a challenge now that welfare science thinking has advanced to the point where it is widely considered that evaluation of positive, as well as negative, mental state should be an integral component of animal welfare assessment [8]. Moreover, for assessment of zoo animals, there is a need for methods to be non-resource intensive and relatively non-invasive. Behavior-based methods are therefore likely to be the most practical to apply.

A variety of welfare assessment models and tools have been derived that include behavioral elements. Commonly used concepts include the Five Domains model [7], and the Welfare Quality^®^ protocol [12]. The former is based on four functional domains of nutrition, environment, health and behavior, and a final cumulative mental state domain. Moreover, the model has recently been updated to expand consideration of the behavioral domain, now named “behavioral interactions”, and focuses on deriving any evidence of animals seeking goals when interacting with the environment or other species, including humans [13]. The Welfare Quality protocol is similarly based on four principles of feeding, housing, health, and appropriate behavior, but expands on these to provide 12 assessment criteria. Both of these models emphasize a focus on animal-based, as opposed to resource-based, measures, with the former being a measure of actual animal welfare and including the effects of the resource inputs supplied [14,15]. These tools have mainly been validated and implemented for assessment of welfare of agricultural animals. However, some work has been performed in adapting and trialing these methods to animals in a zoo environment. For example, Sherwen et al. 2018 carried out an extensive cross-species evaluation of a Five Domain-based assessment protocol [16], whilst Clegg et al. 2015 and Salas et al. 2018 adapted the Welfare Quality Framework for bottlenose dolphins (*Tursiops truncatus*) [17] and Dorcas gazelles (*Gazella dorcas*) [18]. A recent framework that considers the somewhat unique situation of conservation breeding, where welfare strategies need to consider the likelihood of successful reintroductions to the wild, as well as reducing incidence of stress and adverse health is the Opportunities to Thrive program [19]. This assessment schema has a focus on positive indicators of welfare, rather than the absence of negative indicators, achieved by an evaluation of management inputs, as well as outputs in terms of health and behavioral expression.

In comparison to mammals, signs of pain or distress are less well understood for reptiles, and their behavioral repertoire under-researched [20]. Furthermore, due to their ectothermic physiology with low metabolic rate, they are often considered less behaviorally expressive than mammals [15]. As a result, assessment of welfare is likely to be more difficult than in other zoo species. In spite of this, the use of animal-based measures to assess welfare is likely to offer similar advantages as in their mammalian counterparts, being a more direct measure of their experience. Therefore, there is a need to identify reptile-specific, animal-based, indicators of affective state, and use these to derive reptile-specific welfare assessment tools for implementation in zoos.

In spite of substantially less focus being directed towards reptile welfare in comparison with mammals, there is an expanding literature base, as well as increasing recognition based on this literature base, of their ability to experience feelings, and thus be recognized as sentient [21,22]. A number of studies have evaluated methods of improving welfare in reptiles, as well as use of spontaneously observed animal-based indicators of a changed welfare state. A number of these studies have examined the behavioral responses to provision of enrichment, as a method assumed to improve animal welfare. The occurrence of abnormal repetitive behaviors, such as escape behaviors [23], increased locomotor exploration, [24,25,26], increased foraging behaviors [24,25,26], behavioral response to novelty [25], along with increased visibility and loose-coiling whilst resting in snakes [27], have all been proposed as useful methods to ascertain welfare. Other behaviors suggested as indicators of stress in reptiles include body inflation and hissing, aggression directed towards conspecifics and humans, and interaction with transparent boundaries [28,29,30]. A challenge with incorporating some of these into a welfare assessment tool, used as a snapshot at a point in time, is that they may need contrast with an alternately housed group to assess relative incidence or require temporal determination. Therefore, there is a need to develop guidance for indicators for welfare assessment which considers the practical needs of users. Since there are a significant number of welfare indicators already identified in the literature, it would be useful to refine this list for further practical validation. Ascertaining opinion from experts is one method to further define and develop this list of indicators.

Use of expert opinion, through employing methods such as surveys or focus groups, can be an expeditious way of gathering information, especially where there is limited literature [31]. The Delphi technique, named after the Greek oracle, is a method used to enable a group to communicate on an issue through a formalized communication process. The technique consists of two or more rounds of questionnaires or interviews (usually aimed at experts), with summarized responses feeding into the next round of questions [32]. Versions of the Delphi consensus method have been used for identifying welfare indicators in elephants [33], mice [34], horses [31], rabbits [35], and tigers [36], amongst others. The aim of this study was to identify, through expert consensus, animal-based indicators of welfare for four families of reptiles commonly housed in captivity. Families considered were Agamidae, Chelidae, Pythonidae, and Testudinidae. A focus was on identifying indicators that were both practical and valid for use in a daily welfare assessment and longer audit-type assessment of reptiles housed in captivity.

## 2. Materials and Methods

### 2.1. Ethical Statement

This study was conducted with approval from the University of Adelaide’s Human Research Ethics Committee (H-2020-077). Informed consent was obtained from all participants in each survey, including the pilot surveys, via the online survey platform.

### 2.2. Study Objectives

The Delphi consultation method was used to determine, through expert opinion, the most valid (defined as how accurately the criterion indicates animal welfare status) and practical indicators to assess welfare, as well as the valence of affective state inferred from these indicators, for four taxonomic families: Agamidae, Chelidae, Pythonidae, and Testudinidae. Experts were defined as those with experience in the captive management and/or veterinary care of reptiles, or in welfare science in relation to reptiles. When targeting respondents, we considered that an expert would be spending a majority of their time working with reptiles, e.g., a herpetology researcher or zoo keeper with responsibility for reptile exhibits. Indicators suggested were grouped based on their utility as part of an audit or a more simple/rapid daily check. An audit was defined as a benchmarking approach drawing on extended observations over time, which may not be practical to perform as part of a daily check and may involve increased animal handling, extended observation, or detailed veterinary assessment. A daily check was defined as being able to be easily performed based on observations of the animal while doing routine husbandry tasks (e.g., cleaning, feeding), such that viewing of the animal was for a short time with minimal handling opportunity. This study was conducted in four phases, which are outlined in Figure 1.

### 2.3. Study Phases

#### 2.3.1. Survey Development

Investigative meetings were conducted over 3 weeks in April 2020. During these, the study team evaluated literature; see, e.g., [28,29,30], and consulted with local colleagues at Zoos South Australia to derive a list of potential indicators of welfare for the four families of reptiles. There was a focus on identification of animal-based, rather than input-based, measures, since the latter are already in use and have received comparatively more attention. In selecting behaviors, we asked respondents to make the assumption that all resource-based measures were being met (Table 1).

In deriving the criteria, the four physical/functional domains of the Five Domains [7] were used to provide indicative welfare principles, being nutrition, environment, health, and behavior, with these principles being sub-divided into nine welfare criteria, based upon the Welfare Quality^®^ Principles and Criteria (Table 2) [12]. Proposed behavioral indicators were presented under the appropriate Welfare Criterion. The fifth domain, mental state, was taken into consideration by asking respondents to indicate valence of affective state inferred by the assessment criterion proposed.

The Qualtrics^®^ (https://www.qualtrics.com/) platform was used for all surveys. Following the early consultation and development phase, a pilot survey was produced and remained live for 1 week. Twelve participants, known locally and with experience in reptile husbandry, survey design, or animal welfare, were invited to complete the questions to assess question clarity, survey useability, and time taken to complete. These results were used to refine the survey for the first round of the Delphi consultation.

#### 2.3.2. Delphi Survey 1 

Participants were contacted through a range of channels, including personal contacts (including experienced reptile breeders), the Australasian Zoo and Aquarium Association, professional groups including the Association of Reptile and Amphibian Veterinarians and the European College of Zoological Medicine, and academic researchers who had published on reptile behavior/welfare. Personal emails asking for participation in this study were sent to approximately 150 experts that we identified through our networks. Snowball sampling was allowed by participant recruitment of their colleagues and through the use of Listservs. Consenting participants were informed of the aim, methodology, and duration of this study via an information sheet attached to the survey. This survey was ‘active’ for three weeks (July 2020).

The survey preamble contained information on the welfare principles of the Five Domains and the Welfare Quality framework, as well as definitions for validity, practicality and affective state. The survey commenced with demographic questions, which included questions on age, gender, occupation, education, and years of experience with reptiles. A total of 55 indicators were included in Delphi Survey 1 for participant assessment. Behavioral indicators included those related to feeding, utilization of environment, social interactions, and breeding behavior. Respondents were then asked to rate each assessment indicator for its validity and practicality as an indicator of welfare in the taxonomic family specified, using a score from 1–5 (highly invalid/impractical-highly valid/practical). Additional options were provided to allow respondents to assess indicators as not applicable to a taxon, or to answer ‘Don’t Know’ (Figure 2). Respondents were also asked (where relevant) whether each indicator was expected to infer a positive, neutral or negative affective state. Free-text boxes were provided for further comments or clarification of response. The final section of the survey examined behavioral indicators we considered to be less well established. These may have only received passing referral in the academic literature, or have been mentioned to us by people in our networks with reptile experience. Respondents were first asked whether they had observed the behavior referred to, and then directed towards an assessment of its validity and practicality.

#### 2.3.3. Delphi Survey 2

Delphi Survey 2 was a synthesised version of Survey 1, where the results and questions were reformulated and condensed to allow the same participants to re-evaluate and/or clarify their previously provided answers. Based on the validity and practicality results from Delphi Survey 1, a ranked list of assessment criteria was created. This was achieved by ordering the indicators based on the mean of the respondents’ ratings for that indicator and only presenting indicators that had a 70% consensus and had a mean rating at 4 or above for validity or practicality. Consensus was determined when 70% of the respondents rated the indicator at either a 4 or 5, i.e., valid or practical. The survey was again piloted by local contacts to check for formatting and comprehension concerns. In Delphi Survey 2, participants were asked whether they agreed or disagreed with the rank order of indicators presented in the refined list based on their validity and practicality. If they disagreed, they were asked to reorder the indicators into a rank position they considered more suitable, and to remove any indicators considered invalid or impractical. Finally, participants were asked to select their top five indicators for a daily welfare check. The survey was ‘active’ for two weeks.

### 2.4. Data Analysis

Delphi survey methodology is a qualitative method for collecting information from expert opinion [34]. Data collected from the participants’ responses regarding the validity, practicality, and valence of affective state of each of the behavioral indicators were analysed using descriptive statistics. A 70% consensus was required for an indicator to be considered for presentation in Delphi Survey 2. This percentage was derived from previous research where Delphi consultations had been utilized [34,37].

Fisher’s Exact Tests were performed to determine whether there were associations between occupation or years of experience with rating of an indicator as either valid or practical in Delphi Survey 2. A Bonferroni correction was applied to account for multiple comparisons. All analyses were performed in IBM SPSS (SPSS Inc., Chicago, IL, USA) statistical software.

## 3. Results

### 3.1. Demographics

Delphi Survey 1 received a total of 104 responses. Delphi Survey 2 had 23 responses. Of the final 23 participants that completed both rounds of the Delphi consultation, 19% (6) were zookeepers, 10% (3) were animal welfare officers, 19% (6) were researchers, 19% (6) were veterinarians, 13% (4) were breeders, and 19% (6) listed ‘other’ as their occupation, with 5 respondents occupying multiple roles. Those listing ‘other’ variably described their occupation as science communicator, wildlife sanctuary owner/operator, retail pet store employee, and hobby reptile keeper. Most participants had some form of formal education: bachelor’s degree (32%), vocational education and training (26%), doctorate degree (21%), veterinary qualification (16%), and honours degree (5%), with a number having multiple qualifications. A majority of those that answered Delphi Survey 2 were highly experienced with reptiles with 61% having over 10 years’ experience (22% (5), 0–5; 9% (2), 5–10; 22% (5), 10–15; 9% (2), 15–20; 30% (7), 20+ years; 9% (2), unanswered).

### 3.2. Indicators Identified

Indicators that were rated at 4 or greater for validity and/or practicality in Delphi Survey 1, and that achieved a 70% consensus from the participants, progressed to Delphi Survey 2. Out of the original 55 indicators presented, 36 progressed through to Delphi Survey 2 across all the families considered. Indicators across all taxa that progressed to Delphi Survey 2, with their definitions, are presented in Table 3. A list of behaviors excluded after Delphi Survey 1 is presented in the Appendix A (Table A1). There was considerable variability in the responses provided regarding the valence of affective state that behaviors might indicate. These responses included some non-responses, some multiple responses, and answers of ‘unknown’. The descriptor listed by the majority of respondents across the four taxa has been provided in Table 3. These categorisations have not been analysed formally with statistics and care should be taken in interpretation. Based on this descriptive statistical analysis, the valence of affective state inferred by the indicators was reported to be positive for 12/34 indicators (35%), negative for 16/34 (47%), and neutral for 6/34 (18%).

Ranking of indicators based on both validity and practicality for each family after Delphi Survey 2 is provided in Figure 3, Figure 4, Figure 5 and Figure 6. Indicators are ordered based on the mean of the ranks given by survey participants. Despite only progressing those indicators which achieved 70% consensus in Delphi Survey 1 through to Delphi Survey 2, some Delphi Survey 2 respondents categorized them as either invalid or impractical. Therefore, consensus (defined based on whether respondents kept the indicator in the list or not) for validity or practicality is also presented in Figure 3, Figure 4, Figure 5 and Figure 6. These results were based on participants’ consideration of the validity and practicality of indicators for an audit assessment.

Based on Figure 3, Figure 4, Figure 5 and Figure 6, it can be seen that nine indicators achieved consensus for both validity and practicality for Agamidae: wounds, lameness, species-specific interest and alertness, skin quality, body condition score, physical damage, normal respiration, normal species-specific feeding, and co-occupant aggression (Figure 3).

Eight indicators achieved consensus for both validity and practicality for Chelidae: oculo-nasal discharge, wounds, normal species-specific feeding, skin quality, physical damage, co-occupant aggression, relaxed body movements and locomotion, and deformities (Figure 4).

Nine indicators achieved consensus for Pythonidae: oculo-nasal discharge, wounds, species-specific interest and alertness, normal respiration, body condition score, relaxed body movements and locomotion, normal species-specific feeding, co-occupant aggression, and physical damage (Figure 5).

Ten indicators achieved consensus for Testudinidae: oculo-nasal discharge, lameness, wounds, species-specific interest and alertness, co-occupant aggression, physical damage, normal species-specific feeding, relaxed body movements and locomotion, skin quality, and deformities (Figure 6).

### 3.3. Top Five Indicators for a Daily Check

Participants were asked to select their top five indicators for a daily check of each family based on both validity and practicality, i.e., overall utility. These indicators are presented in Table 4.

No associations between occupation or years of experience and selection of any of the indicators were observed when tested using Fisher’s Exact Tests, and a Bonferroni correction.

## 4. Discussion

This study has generated a list of potential welfare indicators, based on expert opinion, for the four families of reptiles that can go on to be trialed to validate their utility as part of a welfare assessment tool. Given the paucity of information available on animal-based indicators in reptiles, this represents an important step in the development of taxon-tailored welfare assessment tools for this understudied group of animals.

### 4.1. Welfare Indicators Identified

Indicators that emerged as both valid and practical for assessment across the families were largely similar, being measures such as presence of wounds or oculo-nasal discharge, normal feeding, interest and alertness, presence of co-occupant aggression, and physical damage. These indicators are of the type that are utilized widely in welfare assessment tools across many domesticated species, and are in many ways focused around health. The consensus selection of these types of indicators was contrary to our intent in this project to develop a list of behavioral-based indicators, given the benefits of behavior-based measures as welfare indicators and that reptiles are often considered behaviorally inexpressive. A number of behavior-based indicators did progress through to our stage 2 survey (for example, interaction with transparent boundaries), but these often failed to reach consensus on practicality. Behaviors associated with breeding which were originally selected for inclusion in the survey universally failed to make it into the stage 2 survey. Although mating behaviors are ‘natural behaviors’ which typically infer a positive affective state [38], these behaviors can only be observed during breeding seasons and are not practical throughout the remaining times of the year. Additionally, these are not valid indicators if the captive reptiles are sexually immature, or if they are maintained in single-sex or individual housing. It is also of note that the list of eight to ten indicators that achieved consensus for validity and practicality for each reptile family for audit purposes bears remarkable similarity to the top five selected for a daily check. These are generally easy to assess, being visibly obvious and non-transient measures, indicative of health. It is worth noting that body condition scoring did not achieve consensus in Chelidae and Testudinidae, or make the top 5 daily check list. Free-text comments provided some insight into this reasoning: that body condition is challenging to evaluate accurately due to the shell and the difficulty in visually determining variation in muscle mass on their distal forelimbs and sagittal crest [39]. It has previously been suggested that in these shelled species, condition score may be better used for evaluating a population across time, rather than as a snapshot indicator [40].

The selection of these types of traditional health-based indicators may have arisen for two reasons. Firstly, that veterinarians and animal caretakers are more conditioned to using these types of assessment indicators and may therefore be more likely to have favored them in this survey than consider novel indicators. Secondly, that health-based indicators may actually be the most valid and practical indicators of welfare in these species. A challenge with assessing reptile welfare particularly is that animals may not be visible, being less gregarious or more cryptic. Therefore, subtle, transient behaviors are likely to be missed without intensive observational effort and would likely not be practical to include in a welfare assessment tool. It has also been suggested that reptilian behavior is not as ‘intuitive’ as mammalian behavior, leading to misinterpretation of behavior-based indices [41]. Another consideration is that, as ectotherms, reptilian behavior is strongly influenced by the environment. As environmental conditions change, behavior will also change, and this may have influenced thinking in the survey responses [42]. Furthermore, since handling is likely to be stressful, overt signs that can be seen from a distance are likely to be preferable for inclusion in a tool.

It is now well recognized that whilst health is a component of good welfare, it is but one facet. The identification of positive welfare indicators represents a pro-active approach, more in line with societal thinking, and an increasing shift in legislative direction to not just provide conditions that allow attainment of an acceptable level of animal welfare, but allow the development of high welfare systems [8,43]. Furthermore, positive emotional state is an established component of common welfare assessment models and tools, such as the Five Domains and Welfare Quality Protocols [7,12]. In the absence of a body of literature where there has been dedicated study and confirmation of the types of behaviors that represent positive emotions in reptiles, the identification of indicators of positive emotions is likely to be performed by analogy with mammalian counterparts [15]. Behaviors that are likely candidates are social affiliative and play behaviors [15], positive human–animal interactions that prioritize consideration of what the animal wants from the interaction [44,45], the expression of goal-directed or anticipatory behaviors [46,47], and the use of qualitative behavioral assessment, where the expressive body language of animals is assessed [48].

The survey results showed significant variability and discrepancy between experts’ beliefs as to the valence of affective state inferred by the indicators. This resulted in a determination that the majority of indicators inferred a negative affective state, with around a third implying a positive affective state. However, it is posited that respondents may have considered behaviors as intrinsically positive or negative, and this may not reflect an accompanying valence of affective state. For example, normal respiration was deemed by the majority to be indicative of positive affective state. Whilst this is a positive sign of health, it represents normal physiology and arguably does not imply that the animal actually has a positive mental state. Such a misinterpretation of common behavioral and biological signs as positive indicators of welfare has been observed previously by Warwick et al. 2013 [28]. Given the variability in responses, as well as possible miscomprehension, care should be taken in interpreting this categorization without further investigation. At the current time, it seems clear that in reptiles, as in many mammals, there is lack of identification and consensus as to which behaviors might indicate positive emotional states. This renders the goal of creating a welfare assessment tool, with a broad range of positive and negative descriptors, currently challenging. There clearly needs to be more focused research on this topic utilizing methods such as judgement bias [46,49], or preference testing [50], to corroborate determinations on positive emotions associated with behaviors identified. Alternately, the employment of habitat modifications expected to be positive, with evaluation of a suite of indicators before and after the change might be an alternate validation strategy; see, e.g., [24]. In spite of the information provided in the preamble to the survey around the welfare frameworks that the survey was based on, it may also be beneficial in future surveys of this nature to gauge the respondents’ understanding of animal welfare at the start of the survey. This may perhaps be achieved through use of open-ended questions with subsequent textual analysis.

Whilst it is relatively easy for someone who works regularly with a group of animals to be able to assess practicality based on their experience, assessment of validity is much more challenging. We defined validity as how well the indicator reflects animal welfare. However, assessment of this ideally requires other corroborating data obtained at the time the behavior was being performed. This might include physiological data such as corticosterone or other biochemical markers, or use of other behavior-based assessment means, as previously discussed. [15] In the absence of this corroborating evidence, validity can potentially be determined based on context of the behavior and its association with other events occurring at the time the behavior was performed. This is likely open to interpretation and not necessarily accurate-perhaps explaining the inability to achieve consensus on such indicators. However, the experience level of the experts in the second round was substantial, and it is considered that this does enhance the reliability of the findings, despite the previous discussion.

Assessment of animal welfare in zoos and other captive settings is certainly a growing focus in both the scientific literature and across captive animal industry husbandry/care guidelines [3,51]. Such assessments typically focus on either an institutional level [16], or a more species-specific approach that is considered a more extensive assessment of an individual’s welfare at a point in time (e.g., [17,52]). These species-specific assessments can also vary in the frequency of use with some researchers and animal caretakers incorporating structured, rapid welfare assessments into daily or weekly checks of an animal. This approach is designed to monitor any changes in an individual’s likely welfare state over time to potentially allow for early intervention if indicators deviate from ‘normal’ [53]. Given the lack of information currently available on captive reptile welfare, this study sought to identify indicators for potential use in a range of welfare assessment protocols-both longer audit-style assessments and a more rapid, user-friendly ‘daily check’ assessment as outlined in Table 4. Future research should be dedicated to incorporating both the ‘daily check’ and audit-style welfare indicators into husbandry and management to track scores over time, and, ultimately, evaluate the effectiveness of this approach to support welfare improvement in this group of animals.

### 4.2. Delphi Methodology and Study Limitations

Delphi consultation methods have been considered effective ways of gauging expert opinion, which are particularly valuable where an evidence base is lacking [31]. However, there may be concerns about repeatability, and whether another group of experts would arrive at the same determination [31]. We did find some evidence supporting this concern, such as apparent inconsistency in responses from Survey 1 to Survey 2. After Survey 1, only those indicators that achieved a 70% consensus for validity and practicality were taken through to Survey 2. However, we offered the option for respondents to categorize these as invalid or impractical in Survey 2 and this option was taken in a number of cases, despite the previously established consensus. This action by respondents, combined with the reduced number of respondents in Delphi Survey 2, likely led to the somewhat counter-intuitive finding that many of the top-ranked indicators actually failed to achieve consensus (see Figure 3, Figure 4, Figure 5 and Figure 6). This has arisen through a relatively small number of respondents ranking these indicators highly and positioning them at the top of the lists for validity/practicality. However, a significant number of respondents have rejected the indicators inclusion in the list causing the failure to achieve consensus, as well as leading to a relatively skewing since a mean rank is influenced not only by position in the list but the number of respondents including it in the list. An exclusion from the list reduces the denominator in calculation of the mean ranks. This finding highlights the importance of considering both consensus as well as strength of opinion, as exemplified in the Delphi approach. Some survey design issues may have contributed to this finding. With the inclusion of 55 behaviors and four families of reptiles in Survey 1, the survey was long and potentially felt repetitive. Furthermore, the question format considered the four families in parallel in each question. Taken together, these factors may have introduced elements of ‘survey fatigue’ leading to indiscriminate box ticking. In future work, consideration should be given to splitting taxa into separate surveys, allowing more focused targeting of experts in each taxon, as well as other considerations to optimize survey structure and length. There is also the possibility that due to the large number of species within each family considered, differences in behavioral repertoire are likely, which may hinder achievement of consensus. Whilst development of genus or species-specific welfare assessment tools would be the ideal, these may not be practical for zoos to implement, at least for regular use. Furthermore, whilst snowball sampling was only one method for participant recruitment, it is prudent to consider whether this method allows unconscious bias to creep into research findings since respondents may refer on to those of ‘like-minds’. This is likely less of a concern in studies such as this where the population is fairly homogenous in terms of the characteristic of interest, i.e., reptile experience, but worth bearing in mind in future study design. Other Delphi surveys with an animal welfare focus have used experts to determine prioritization of welfare issues; see, e.g., [31,35,54]. For these types of surveys there is no further way to validate the findings, and hence methodological issues are of paramount importance. Given that the indicators that have arisen from the current study will need further validation and testing in a practical scenario these issues are likely of less concern in this type of exploratory, rather than deterministic analysis since behaviors nearing consensus can still be examined in a practical scenario.

There was a significant dropout rate between Delphi Survey 1 and Delphi Survey 2. Dropout in these types of investigations is to be expected, with suggestion that at least 20% is not uncommon [55,56]. However, the rate in the current study was 78% which is disappointing. As a result of the high drop out rate, if the same number of people said a particular measure was not practical in both surveys, the percent of people rating that measure as not practical would be higher in the second survey. This may account for the decrease in consensus seen around measures that previously achieved consensus. This dropout rate may have been influenced by global events at the time of conducting this study, being in the midst of a pandemic, although we were still able to attain good participation in the first survey. It was beneficial to have a good response rate of 104 for the first survey since this allowed the identification, with some certainty, of a smaller subset of indicators to take through to the second survey. Furthermore, the 23 respondents who completed the second survey contributed many years of experience with reptiles and therefore truly provided an expert opinion. Many other animal welfare-focused Delphi surveys have engaged a similar number of experts. For example, in a study on horse welfare, 19 experts were consulted [31], 11 were involved in a rabbit study [35], and 21 experts were consulted on farm animal welfare issues [54].

The format of the consultation as a survey may have also influenced the only moderate consensus that arose on validity/practicality of the indicators in the second survey in spite of the previous consensus. Rioja-Lang et al. 2020 [31] similarly found only moderate consensus between experts arising from online methods examining the nature of horse welfare issues in the UK. In that study, initial findings were followed up with face-to-face discussions where better consensus was achieved. The characteristic Delphi method is anonymous to reduce biasing of responses and intimidation of others [35]. Survey methodology clearly meets this requirement for anonymity. However, as reasoned in Rioja-Lang et al. 2020 [31], face-to-face discussion promotes breaking down of ideas, encourages debate and reflection and may therefore lead to more valid findings. A hybrid model utilizing anonymous responses to gather preliminary data, with subsequent guided group discussion, may be a preferred model.

## 5. Conclusions

This Delphi consultation has identified a range of indicators that have been considered as both valid and practical for welfare assessment in four families of reptiles: Agamidae, Chelidae, Pythonidae, and Testudinidae. These indicators were largely health related, but some other indicators that are behavioral in nature were also identified. Other behavioral -based indicators almost reached consensus and may be worthy of further investigation. These indicators should be further developed into welfare assessment tools for the various families of reptiles, and validation work performed. This study represents a next step towards developing animal-based welfare indicators for this under-researched group of animals, and thus ultimately contributing to welfare and husbandry improvements for captive reptiles.

## Figures and Tables

**Figure 1 animals-11-02010-f001:**
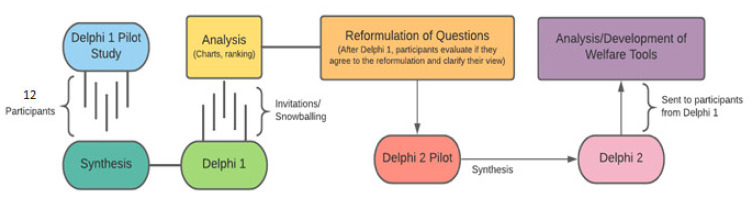
Flowchart depicting the study phases.

**Figure 2 animals-11-02010-f002:**
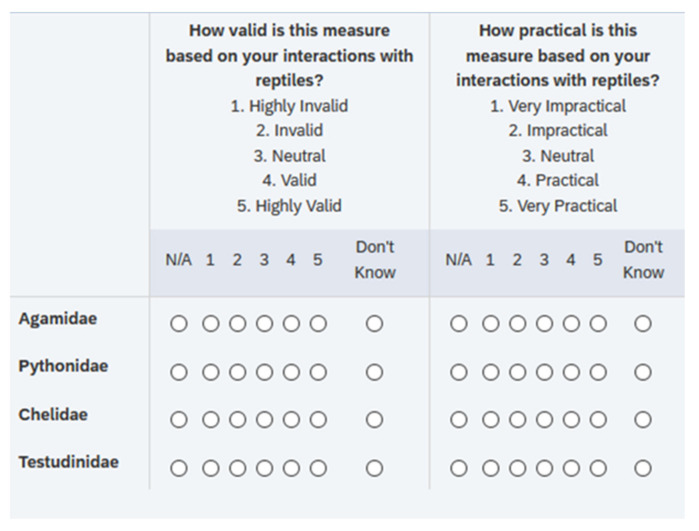
Delphi Survey 1 excerpt illustrating rating system and layout.

**Figure 3 animals-11-02010-f003:**
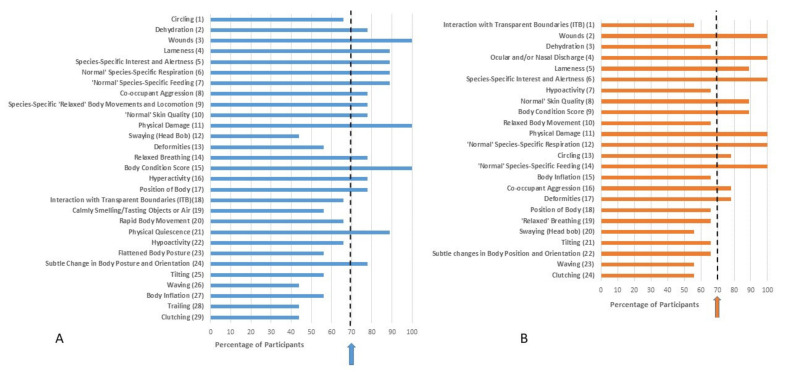
Ranking of indicators from Delphi Survey 2 for Agamidae based on (**A**) validity and (**B**) practicality for an audit welfare assessment. The rank position of the indicators was determined based on the mean of the ranks given by survey participants and is indicated in brackets. Whilst the indicators presented in Delphi Survey 2 were chosen based on high scores for validity/practicality in Delphi Survey 1, some participants in Delphi Survey 2 considered them as either invalid or impractical. Percentage consensus on whether the indicator is considered valid or practical is indicated on the x axis. The arrow represents the 70% consensus level chosen for this Delphi consultation, i.e., 70% of respondents retained the indicator in the list.

**Figure 4 animals-11-02010-f004:**
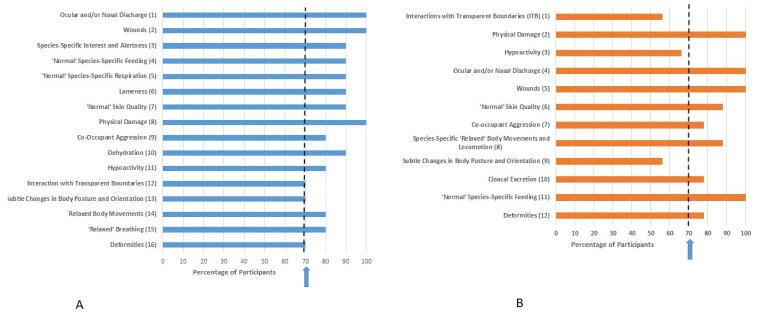
Ranking of indicators from Delphi Survey 2 for Chelidae based on (**A**) validity and (**B**) practicality for an audit assessment. The rank position of the indicators was determined based on the mean of the ranks given by survey participants and is indicated in brackets. Whilst the indicators presented in Delphi Survey 2 were chosen based on high scores for validity/practicality in Delphi Survey 1, some participants in Delphi Survey 2 considered them as either invalid or impractical. Percentage consensus on whether the indicator is considered valid or practical is indicated on the x axis. The arrow represents the 70% consensus level chosen for this Delphi consultation, i.e., 70% of respondents retained the indicator in the list.

**Figure 5 animals-11-02010-f005:**
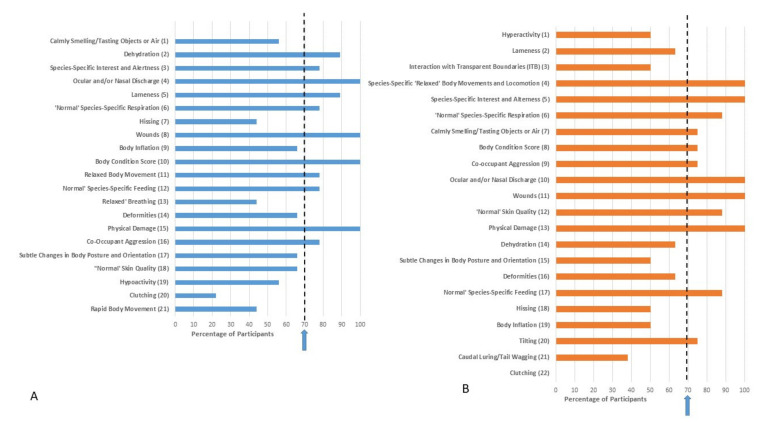
Ranking of indicators from Delphi Survey 2 for Pythonidae based on (**A**) validity and (**B**) practicality for an audit assessment. The rank position of the indicators was determined based on the mean of the ranks given by survey participants and is indicated in brackets. Whilst the indicators presented in Delphi Survey 2 were chosen based on high scores for validity/practicality in Delphi Survey 1, some participants in Delphi Survey 2 considered them as either invalid or impractical. Percentage consensus on whether the indicator is considered valid or practical is indicated on the x axis. The arrow represents the 70% consensus level chosen for this Delphi consultation, i.e., 70% of respondents retained the indicator in the list.

**Figure 6 animals-11-02010-f006:**
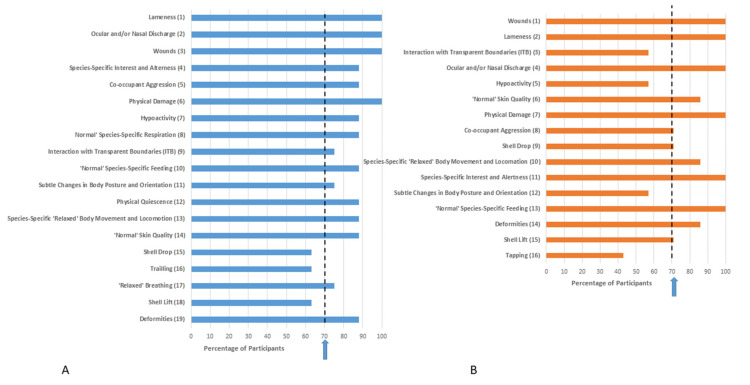
Ranking of indicators from Delphi Survey 2 for Testudinidae based on (**A**) validity and (**B**) practicality for an audit assessment. The rank position of the indicators was determined based on the mean of the ranks given by survey participants and is indicated in brackets. Whilst the indicators presented in Delphi Survey 2 were chosen based on high scores for validity/practicality in Delphi Survey 1, some participants in Delphi Survey 2 considered them as either invalid or impractical. Percentage consensus on whether the indicator is considered valid or practical is indicated on the x axis. The arrow represents the 70% consensus level chosen for this Delphi consultation, i.e., 70% of respondents retained the indicator in the list.

**Table 1 animals-11-02010-t001:** List of general resource-based measures assumed to be met when deriving welfare assessment criteria.

Enclosure	Appropriate: – Enclosure design – Space allowance – Substrate and furnishings
Temperature	Basking gradients Optimum temperature range Natural variation to allow behavioral thermoregulation
Water	Ability to soak Appropriate water provisions for drinking/hydration
Humidity	Optimum humidity range
Lighting	Appropriate photoperiod Mimic the natural outdoor seasonal lighting Exposure to natural sunlight, or a comparable range of wavelengths including UVA (320–400 nm) and UVB (290–320 nm)
Nutrition	Appropriate: – Nutrient composition of the diet – Dietary items (e.g., insects provided to an insectivorous reptile) – Presentation of food in a way that encourages natural foraging behavior
Socialization	Appropriate social housing (e.g., isolated or group housing; mixed or sex-based housing)

**Table 2 animals-11-02010-t002:** Framework for development of proposed assessment criteria based on the Five Domains and Welfare Quality Principles.

Welfare Principal	Welfare Criteria
1. Nutrition	1.1. Good nutrition and hydration
2. Physical Health	2.1 Absence of injuries and disease
	2.2 Absence of pain induced by management procedures
	2.3 Good human–animal relationship
3. Environment	3.1 Thermal/lighting comfort
	3.2 Comfort around resting
	3.3 Ease of movement
4. Behavior	4.1 Expression of social/courting behaviors
	4.2 Other behaviors

**Table 3 animals-11-02010-t003:** Definitions of behaviors which achieved a 70% consensus rate for validity and/or practicality in Delphi Survey 1 and progressed to Delphi Survey 2. The valence of affective state represents the majority response by the participants across all families. The taxa to which the behavior is applicable is derived from the results of Delphi Survey 1 and represents the options for ranking presented in Delphi Survey 2.

Descriptor	Definition	Valence of Affective State	Applicable Taxa
Body Condition Score	A visual assessment of the amount of fat and muscle covering the bones of the animal	Not asked	Agamidae, Pythonidae
Body Inflation	Intentional, and often repetitive, body inflation and deflation	Negative	Agamidae, Pythonidae
Calmly Smelling/Tasting Objects or Air	Flicks tongue, capturing particles in the air	Positive	Agamidae, Pythonidae
Caudal Luring/Tail Wagging	Animal vibrates tail rapidly, moving back and forth on the same plane	Positive	Pythonidae
Circling	Animal pacing around the perimeter of objects or cage mates	Negative	Agamidae
Cloacal Excretion	Defecation and/or urination when handled	Negative	Chelidae
Clutching	Animals gripping at various intensities on handler or object	Not applicable	Agamidae, Pythonidae
Co-Occupant Aggression	Defensive or aggressive biting, chasing, or ramming	Negative	Agamidae, Chelidae, Pythonidae, Testudinidae
Deformities	Malformation of the body	Negative	Agamidae, Chelidae, Pythonidae, Testudinidae
Dehydration	Visible indicators of dehydration	Negative	Agamidae, Chelidae, Pythonidae
Flattened Body Posture	Flattening body against surfaces	Neutral	Agamidae
Hissing	Sound caused by expelling air from the glottic making a small cartilage piece inside the glottis vibrate	Negative	Pythonidae
Hyperactivity	Abnormally high levels of physical activity for the species (often associated with ITB)	Negative	Agamidae, Pythonidae
Hypoactivity	Abnormally low levels of physical activity for the species	Negative	Agamidae, Chelidae, Pythonidae, Testudinidae
Interaction with Transparent Boundaries (ITB)	Persistent attempts to push against, crawl up, dig under/around transparent barriers of the enclosures	Negative	Agamidae, Chelidae, Pythonidae, Testudinidae
Lameness	An abnormal gait/stance of an animal in an attempt to reduce pain. Assumed to relate to ‘slithering’ ability in pythons	Negative	Agamidae, Chelidae, Pythonidae, Testudinidae
‘Normal’ Skin Quality	Skin in good health, with absence of abnormal texture, pigmentation, etc. indicative of active disease process or physiological response to stress	Positive	Agamidae, Chelidae, Pythonidae, Testudinidae
‘Normal’ Species-Specific Feeding	Feeding habits as usual for the species	Positive	Agamidae, Chelidae, Pythonidae, Testudinidae
‘Normal’ Species-Specific Respiration	Normal respiratory rate, absence of abnormal respiratory noises or excessive respiratory effort (e.g., heaving, gasping)	Positive	Agamidae, Chelidae, Pythonidae, Testudinidae
Ocular and Nasal Discharge	A clear or yellow/white discharge from the eyes and/or nostrils	Negative	Agamidae, Chelidae, Pythonidae, Testudinidae
Physical Damage	Damage to the body not necessarily resulting in broken skin, for example grazes and other lesions	Negative	Agamidae, Chelidae, Pythonidae, Testudinidae
Position of Body	The way and manner the reptile has placed their body in an enclosure. If favouring a side or a limb, this may indicate a sign of discomfort	Neutral	Agamidae
Physical Quiescence	Unremarkable species-specific activity, e.g., free from apprehension and fear activities	Neutral	Agamidae, Testudinidae
Rapid Body Movement	Abnormal jerking, locomotion, or jumping	Negative	Agamidae, Pythonidae
‘Relaxed’ Breathing	Unremarkable breathing habits for the species	Positive	Agamidae, Chelidae, Pythonidae, Testudinidae
Shell Drop	Female drops the edge of the shell on the ground during pushing, hooking, or cloacal probing by the male	Neutral	Testudinidae
Shell Lift	Female presents the cloacal area to view, to olfact, or for penile insertion, by extending the hind limbs and thereby lifting the posterior part of the shell	Positive	Testudinidae
Species-Specific ‘Relaxed’ Body Movements and Locomotion	Species-specific environmental exploration	Positive	Agamidae, Chelidae, Pythonidae, Testudinidae
Species-Specific Interest and Alertness	Species-specific ‘relaxed’ interest/awareness in proximate or novel objects, ‘relaxed’ visual/olfactory explorations	Positive	Agamidae, Chelidae, Pythonidae, Testudinidae
Subtle Changes in Body Posture and Orientation	Stretching out of limbs while basking, relaxed adoption of body angles, and using furnishings	Positive	Agamidae, Chelidae, Pythonidae, Testudinidae
Swaying (Head Bob)	Horizontal movement of the head and neck	Neutral	Agamidae
Tapping	Repeatedly lifting the shells and tapping against a co-occupant’s shell	Neutral	Testudinidae
Tilting	Turn on their side	Negative	Agamidae, Pythonidae
Trailing	One animal follows another, typically the male following the female	Positive	Agamidae, Testudinidae
Waving	Animal lifts its forelimbs; feet lateral to its head	Positive	Agamidae,
Wounds	An injury to living tissue caused by a cut, blow, bite, or other form of impact. Typically resulting in broken skin	Negative	Agamidae, Chelidae, Pythonidae, Testudinidae

**Table 4 animals-11-02010-t004:** Top five indicators considering validity and practicality for a daily check. Indicators are ranked based on frequency of responses assigning them to the ‘top five’.

Agamidae	Chelidae	Pythonidae	Testudinidae
Physical Damage	‘Normal’ Species-Specific Feeding	Species-Specific Interest and Alertness	‘Normal’ Species-Specific Feeding
‘Normal’ Species-Specific Feeding	Ocular and/or Nasal Discharge	Normal’ Skin Quality	Ocular and/or Nasal Discharge
Wounds	Physical Damage	Physical Damage	Species-Specific Interest and Alertness Species-Specific ‘Relaxed’ Body Movements and Locomotion Wounds Physical Damage (ranked equally)
Body Condition Score	Wounds	Body Condition Score	
Ocular and/or Nasal Discharge	Species-Specific ‘Relaxed’ Body Movements and Locomotion	Species-Specific ‘Relaxed’ Body Movements and Locomotion Normal’ Species-Specific Respiration Ocular and/or Nasal Discharge (ranked equally)	

## Data Availability

The datasets collected and analysed during the current study are available from the corresponding author upon request.

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
