# Peer review of "Identification of Animal-Based Welfare Indicators in Captive Reptiles: A Delphi Consultation Survey"

_animals, 2021, doi:10.3390/ani11072010_

Round 1

Reviewer 1 Report

This is a well-written manuscript about a well-designed study generating expert consensus around welfare indicators for four families of reptiles. This study is important and novel because it uses the Delphi method to establish consensus around reptile welfare indicators rather than proposing or validating specific measures and because it aims to generate welfare indicators for whole families of reptiles rather than individual species. The manuscript should be publishable in Animals after these minor revisions:

  1. Many studies have already used or validated behavioral welfare indicators for a variety of reptile species, but these studies are not mentioned in the introduction or discussed sufficiently. The authors should
    1. add a paragraph to the introduction reviewing currently-used indicators of reptile welfare, highlighting differences among these indicators, and describing the need to develop consensus. It seems to me this would fit well before Line 104. I would recommend including the sources below:
      • citations 21-23 and 38 from this paper,
      • Bashaw, M. J., Gibson, M. D., Schowe, D. M., & Kucher, A. S. (2016). Does enrichment improve reptile welfare? Leopard geckos (Eublepharis macularius) respond to five types of environmental enrichment. Applied Animal Behaviour Science, 184, 150-160.
      • Hoehfurtner, T., Wilkinson, A., Nagabaskaran, G., & Burman, O. H. (2021). Does the provision of environmental enrichment affect the behaviour and welfare of captive snakes?. Applied Animal Behaviour Science, 239, 105324.
      • Londoño, C., Bartolomé, A., Carazo, P., & Font, E. (2018). Chemosensory enrichment as a simple and effective way to improve the welfare of captive lizards. Ethology, 124(9), 674-683.
      • Moszuti, S. A., Wilkinson, A., & Burman, O. H. (2017). Response to novelty as an indicator of reptile welfare. Applied Animal Behaviour Science, 193, 98-103.
    2. On Line 302, describe their work as an important "next" step rather than an important "first" step.
  2. The authors should add two references that are central to how American animal welfare scientists currently think about welfare:
    1. In paragraph 3, the "Opportunities to Thrive" model, citing Greggor, A. L., Vicino, G. A., Swaisgood, R. R., Fidgett, A., Brenner, D., Kinney, M. E., ... & Lamberski, N. (2018). Animal welfare in conservation breeding: Applications and challenges. Frontiers in Veterinary Science, 5, 323.
    2. On Line 60 and perhaps elsewhere: Whitham, J. C., & Wielebnowski, N. (2013). New directions for zoo animal welfare science. Applied Animal Behaviour Science, 147(3-4), 247-260.
  3. A few methodological details are missing:
    1. In section 2.2.2, the authors should specify the minimum amount or type of experience needed for someone to qualify as an expert and participate in the survey.
    2. In Lines 296-7, the authors should describe how they looked or tested for associations between occupation or years of experience and selection of any indicator.
  4. Two ideas could be added to the discussion:
    1. The high drop-out rate means that if the same number of people said a particular measure was not practical in both surveys, the percent of people rating that measure not practical would be higher in the second survey. How much might the drop-out rate account for the apparent decrease of consensus around some of the measures from Delphi 1 to Delphi 2?
    2. The families of reptiles studied here each include a large number and variety of species with different niches and ecological pressures that might cause behavioral differences. Is it possible more consensus would be observed for behavioral indicators of welfare at the genus or species level?
  5. In Table 3, there's a typo in the definition for "Physical Damage" and the definition of "Shell Drop" would be clearer in active voice: "Female drops the edge of the shell on the ground..." On Line 443, Delphi is misspelled.

Author Response

Many studies have already used or validated behavioral welfare indicators for a variety of reptile species, but these studies are not mentioned in the introduction or discussed sufficiently. The authors should add a paragraph to the introduction reviewing currently-used indicators of reptile welfare, highlighting differences among these indicators, and describing the need to develop consensus. It seems to me this would fit well before Line 104. I would recommend including the sources below:

  • citations 21-23 and 38 from this paper,
  • Bashaw, M. J., Gibson, M. D., Schowe, D. M., & Kucher, A. S. (2016). Does enrichment improve reptile welfare? Leopard geckos (Eublepharis macularius) respond to five types of environmental enrichment. Applied Animal Behaviour Science, 184, 150-160.
  • Hoehfurtner, T., Wilkinson, A., Nagabaskaran, G., & Burman, O. H. (2021). Does the provision of environmental enrichment affect the behaviour and welfare of captive snakes?. Applied Animal Behaviour Science239, 105324.
  • Londoño, C., Bartolomé, A., Carazo, P., & Font, E. (2018). Chemosensory enrichment as a simple and effective way to improve the welfare of captive lizards. Ethology124(9), 674-683.
  • Moszuti, S. A., Wilkinson, A., & Burman, O. H. (2017). Response to novelty as an indicator of reptile welfare. Applied Animal Behaviour Science193, 98-103.

This is a good suggestion and we have briefly listed some behavioural indicators of welfare that have been discussed in these papers in a new paragraph at lines 128-148.

On Line 302, describe their work as an important "next" step rather than an important "first" step.

 Line 302 has been amended to read ‘next’ step rather than ‘first’ step.

The authors should add two references that are central to how American animal welfare scientists currently think about welfare:

    1. In paragraph 3, the "Opportunities to Thrive" model, citing Greggor, A. L., Vicino, G. A., Swaisgood, R. R., Fidgett, A., Brenner, D., Kinney, M. E., ... & Lamberski, N. (2018). Animal welfare in conservation breeding: Applications and challenges. Frontiers in Veterinary Science5, 323.
    1. On Line 60 and perhaps elsewhere: Whitham, J. C., & Wielebnowski, N. (2013). New directions for zoo animal welfare science. Applied Animal Behaviour Science147(3-4), 247-260.

Thankyou for alerting us to these references. Both of these references have been included in the introduction at lines 63-65 and 112-118.

In section 2.2.2, the authors should specify the minimum amount or type of experience needed for someone to qualify as an expert and participate in the survey.

 We have added a note at lines 176-177 that we considered experts would be spending the majority of their time working with reptiles e.g. zookeepers for reptile exhibits or herpetology researchers. All targeted emails asking for participation were to people who we knew satisfied this criteria. There would have been less quality control over those respondents that were recruited through snowballing.  There was no way of screening out in the survey based on time spent with reptiles as we did not engage a panel provider. However, the associations we performed in analysis also showed no differences in responses based on experience levels.

In Lines 296-7, the authors should describe how they looked or tested for associations between occupation or years of experience and selection of any indicator.

We have just added a few words to this section to indicate the statistical tests used to do this.

 Two ideas could be added to the discussion:

    1. The high drop-out rate means that if the same number of people said a particular measure was not practical in both surveys, the percent of people rating that measure not practical would be higher in the second survey. How much might the drop-out rate account for the apparent decrease of consensus around some of the measures from Delphi 1 to Delphi 2?

This is a good point and we have made reference to it at lines 503-506 of the discussion.

  1. The families of reptiles studied here each include a large number and variety of species with different niches and ecological pressures that might cause behavioral differences. Is it possible more consensus would be observed for behavioral indicators of welfare at the genus or species level?

This is an excellent point and we have added some referral to it at lines 488-492. We discussed these issues in designing the study but wanted to make the tool as generic as possible across families to aid the practical implementation of this.

In Table 3, there's a typo in the definition for "Physical Damage" and the definition of "Shell Drop" would be clearer in active voice: "Female drops the edge of the shell on the ground..." On Line 443, Delphi is misspelled.

Thankyou- these mistakes have now been corrected.

Reviewer 2 Report

Animals-1254051

The subject matter of this submission falls within the scope of Animals. However, in its current form, I find it lacking in coherence and I recommend that the authors be asked to conduct a major revision prior to a full review. Given the lack of a working definition of animal welfare at the outset, I am not convinced that the experts held a unified view and so any comparison of valuable welfare indicators that they identified was probably flawed. I detail my concerns below.

Line 73: "entrenched" has a rather negative meaning; I would suggest that the authors use a more neutral adjective, such as "integral" or "essential".

Lines 77-78: The authors identify different approaches of assessing welfare; but they do not state what they actually mean by "animal welfare"; they even state later that the experts may have different views on the meaning of animal welfare. In my opinion, this omission is very important, particularly as they are considering here the welfare of ectothermic animals when most welfare research has been devoted to homeotherms. Perhaps the definition proposed recently (Mormede et al, 2018) is suitably inclusive to be appropriate for both homeotherms and ectotherms:

"The welfare of an animal is the positive mental and physical state related to the satisfaction of its physiological and behavioural needs, as well as its expectations. This state varies according to the perception of the situation by the animal.

Surely, only by declaring what one means by welfare can one be able to judge how best to assess it?

Mormede, P., Boisseau-Sowinski, L., Chiron, J., Diederich, C., Eddison, J., Guichet, J.-L., Le Neindre, P. and Meunier-Saläun, M.-C. 2018. Bien-être animal : contexte, définition et évaluation INRA Productions Animales 31: 145-162.

Lines 87-88: "with the former arguably being a more accurate reflection of animal affective state." I think that the authors could be a little more circumspect here; after all, they cite only one reference in favour of this position.

Lines 98-103: Perhaps the authors could integrate the findings of the review by Lambert et al (2019) that reported in Animals on the variety of evidence (30+ publications) for reptile sentience that already exists. I found this paper after less than five minutes searching in Web of Science.

Lambert, H., Carder, G. and D'Cruze, N. 2019. Given the Cold Shoulder: A Review of the Scientific Literature for Evidence of Reptile Sentience. Animals 9:

Lines 119-121: This is an ethical statement; Lines 122-135: describe a rationale for the Methods and should not be described under the sub-heading Ethical Statement.

I think that the authors need to re-consider the structure of the Methods section in its entirety.

Lines 141-143: See my earlier comment re Lambert et al (2019).

Lines 146-147: "... the assumption was made ..." With what level of certainty are you able to make this assumption?

Table 1: "natural outdoor seasonal lighting" I presume this relates to the patterns of light in the natural habitat and not the location of the collection (e.g. a tropical forest rather than in an high latitude zoo)? Also, what about animals born in captivity and only having experienced the captive environment? This element requires greater precision.

Lines 170-171: "Snowball sampling": to what extent does this type of sampling lead to the inclusion of data from like-minded individuals and the exclusion of  those who might have different, but no-less relevant, views? Have the authors taken this potential bias into account?

Lines 170-171: "Snowball sampling": to what extent does this type of sampling lead to the inclusion of data from like-minded individuals and the exclusion of  those who might have different, but no-less relevant, views? Have the authors taken this potential bias into account?

Lines 184-185: "The final ... well-established." What sort of welfare indicators did the authors regard as less well-established and why?

Without an answer to this question, nobody could repeat this work based upon the Methods as described; that that is the key criterion for the acceptability of a Methods section.

Line 209: "[18] [20]" Should be "[18, 20]". The same error occurs elsewhere in the manuscript.

Lines 218-220: Given the small number of respondents, I think that the authors might also report the actual number of each category of respondent.

Similarly, I am surprised that the authors did not make any effort to gauge the respondents' understanding of animal welfare, a priori. For example, there is still a view held that welfare is all or principally about physical health that does not include the mental state; this view is slowly being changed, but it would be reassuring if the perspective of the respondents was clarified, particularly as some are self-selecting. The definition of Mormede et al (2018) includes mental, physiological and behavioural needs.

Table 3: "Interaction with Transparent Boundaries" is this what is referred to earlier as ITB (see Hyperactivity)? If so, include the abbreviation after "Interaction with Transparent Boundaries" to facilitate comprehension.

Line 302: Given the absence of the Lambert et al (2019) review, I am not convinced that this article is the "first step" (see a similar comment re the Conclusions".

Lines 308-309: Yes, these behaviours are frequently utilised in assessing farm animal welfare; but they are not the only ones. In my opinion, the authors have expressed this thought rather crudely that might lead a reader to understand this text as meaning that these are the only types of indicator used to assess farm livestock welfare, and that is not correct.

Line 310: "health". Given my comment at 218-220, the authors have a similar thought to me, and so some clarification of how the respondents defined welfare would have been useful in the contextualisation of their views.

Lines 357-358: But you have just stated that "... handling is likely to be stressful ..." (Line 345), so what do you mean by "human-animal interactions"? Some clarification is required.

Lines 361-368: Another possible reason for the discrepancy might also be the understanding of what is meant be welfare on the part of the respondents (as mentioned earlier by the authors, Lines ), specifically relating to health and ignoring the mental or affective elements.

Line 364: "posed" I think that the authors probably mean "posited".

Lines 376-377: The authors have presented some useful reptile-focused references relating to the applications of these methods. In my view, in the light of limited experimental work on reptiles, it could be useful to include also some key papers of a more fundamental nature so that readers who might wish to pursue these experimental paradigms have key work identified.

Lines 384-385: Here, the authors are exclusively equating welfare with "a positive or negative emotional state"; this is to ignore elements of physical health that are not linked with emotion (see my earlier comments that refer to the definition proposed by Mormede et al). Are the authors discounting all negative aspects of an animal's state that do not affect its emotion at the moment of assessment? This question returns to my earlier point: how do authors define welfare? and, how do the respondents define welfare? Without that common basis, I think that there are inevitable difficulties of interpretation.

The authors point to some of these challenges; but these are not new (Mason & Mendl, 1993) and have been addressed by numerous authors (e.g. Boissy, Mason, Mendl, Paul) in various publications since then.

Mason, G. and Mendl, M. 1993. Why is there no simple way of measuring animal welfare? Animal Welfare 2: 301-319.

Line 461: As I have indicated at several points, I have significant concerns that the respondents did not share a common view of what animal welfare actually is, and so their identification of indicators that might be useful in assessing welfare is put into doubt.

Line 464: Poor English: "... behavioral-based indicators almost reached consensus..." Should be: ... behavior-based indicators almost reached consensus..."

Line 467: As mentioned earlier, other work has already addressed sentience in reptiles, and so I am not convinced that this study represents the "first step".

Line 512: "C-well" What is this?

Overall, I remain unconvinced of the value of this paper; but I am willing to consider a thorough revision if the editors feel that is appropriate.

Author Response

Line 73: "entrenched" has a rather negative meaning; I would suggest that the authors use a more neutral adjective, such as "integral" or "essential".

The wording has been amended to ‘integral’ here.

Lines 77-78: The authors identify different approaches of assessing welfare; but they do not state what they actually mean by "animal welfare"; they even state later that the experts may have different views on the meaning of animal welfare. In my opinion, this omission is very important, particularly as they are considering here the welfare of ectothermic animals when most welfare research has been devoted to homeotherms. Perhaps the definition proposed recently (Mormede et al, 2018) is suitably inclusive to be appropriate for both homeotherms and ectotherms:

"The welfare of an animal is the positive mental and physical state related to the satisfaction of its physiological and behavioural needs, as well as its expectations. This state varies according to the perception of the situation by the animal.

Surely, only by declaring what one means by welfare can one be able to judge how best to assess it?

Mormede, P., Boisseau-Sowinski, L., Chiron, J., Diederich, C., Eddison, J., Guichet, J.-L., Le Neindre, P. and Meunier-Saläun, M.-C. 2018. Bien-être animal : contexte, définition et évaluation INRA Productions Animales 31: 145-162.

A paragraph defining welfare and the relation of animal welfare with affective states has been added at lines 67-81. This welfare definition has been used as it certainly is appropriate in this case, however, we are unable to access an English translation of this article.  We have alternately access a related webpage source where an English translation has been used to ensure we are reflecting the meaning accurately.

Lines 87-88: "with the former arguably being a more accurate reflection of animal affective state." I think that the authors could be a little more circumspect here; after all, they cite only one reference in favour of this position.

Thankyou for highlighting this sentence. Of course, on reading we realised that the sentence is actually incorrect since animal-based measures are the only way of actually assessing animal welfare, resource-based measures may be used to infer welfare state based on evidence or assumptions as to the effect of the resource on animal affective state. An extra reference to this assertion has been included.

Lines 98-103: Perhaps the authors could integrate the findings of the review by Lambert et al (2019) that reported in Animals on the variety of evidence (30+ publications) for reptile sentience that already exists. I found this paper after less than five minutes searching in Web of Science.

Lambert, H., Carder, G. and D'Cruze, N. 2019. Given the Cold Shoulder: A Review of the Scientific Literature for Evidence of Reptile Sentience. Animals 9:

We are aware of this paper and other papers on reptile sentience. This is a great review that has as its focus a scope of the available literature evaluating various affective states in reptiles. It clearly shows that there has been a wide variety of research on reptiles, how the volume of this contrasts with mammals, the increase in focus over time, and the relative research attention directed towards different orders. It does not however go into detail on the behaviours/indicators described or assessed in the papers that are included in the review, and hence could not inform our specific study design without accessing individual papers referenced within it.  We have however made reference to the review based on some of the findings of the review in terms of scoping the volume of literature available.  This referral is at lines 128-131.

Lines 119-121: This is an ethical statement; Lines 122-135: describe a rationale for the Methods and should not be described under the sub-heading Ethical Statement.

Thankyou for highlighting this error. The ethical statement now forms the first section of the methods. The pre-amble to the study now has a separate heading of ‘study objectives’.

I think that the authors need to re-consider the structure of the Methods section in its entirety.

 We are comfortable with the flow of the methods section, and other reviewers seem to have understood the flow, which at least to our mind describes the steps taken in the order they were performed. We would need some specific guidance on why they are unclear.

Lines 141-143: See my earlier comment re Lambert et al (2019).

We have included reference to Lambert et al in the introduction. As discussed previously, this review provides a general overview of the scope of the literature rather than detailed guidance on behavioural indicators indicative of affective state and hence it is not appropriate to reference it at this point as no indicators were contributed to the survey from this review.

Lines 146-147: "... the assumption was made ..." With what level of certainty are you able to make this assumption?

We are not clear how elaborating on this adds value to the paper. The statement refers to the idea that respondents in answering the survey should assume that resource-based needs are being met when considering animal-based indicators that may be of value. Our survey focus was not to investigate resource-based indicators; hence we did not include questions on this. Since this is posing a hypothetical situation an assessment of certainty is not necessary. We have tried to clarify our meaning here by rewording, so the sentence now reads: ‘In selecting behaviors, we asked respondents to make the assumption that all resource-based measures were being met (Table 1)’.

Table 1: "natural outdoor seasonal lighting" I presume this relates to the patterns of light in the natural habitat and not the location of the collection (e.g. a tropical forest rather than in an high latitude zoo)? Also, what about animals born in captivity and only having experienced the captive environment? This element requires greater precision.

The respondents were asked to consider the situation in relation to animals in captivity. However, they were not asked to consider resource-based measures so we are not clear that elaboration on this is necessary. Clearly, different zoos will have different environments but it would be expected that they would be delivering light intensities/durations comparable with that animals will experience in nature, and that are compatible with their health. It is difficult to be prescriptive around this given the range of families considered and individual variations between zoos.

Lines 170-171: "Snowball sampling": to what extent does this type of sampling lead to the inclusion of data from like-minded individuals and the exclusion of  those who might have different, but no-less relevant, views? Have the authors taken this potential bias into account?

Snowball sampling is often used in qualitative research when the population to be sampled i.e. those with reptile experience, is smaller or less accessible. We believe in this case the target population is sufficiently homogenous in terms of the characteristic of interest (reptile experience) to not cause biasing of the results. Given we weren’t trying to get a random sample it is a valid method to use and it was not the only method of sourcing respondents; it was just not a disallowed method of recruitment. Accordingly, we do not believe its use should have biased the findings.

Lines 184-185: "The final ... well-established." What sort of welfare indicators did the authors regard as less well-established and why?

We regarded them as less-well established since there may have only been passing referral to them in the academic literature, or they may have been mentioned to us by people in our networks with reptile experience. We don’t think this needs elaboration in the manuscript since it was based on our judgement rather than a formal delineation between established and less well-established indicators.

Without an answer to this question, nobody could repeat this work based upon the Methods as described; that that is the key criterion for the acceptability of a Methods section.

Respectfully, we disagree. The survey methods can be replicated based on the information presented here. All the behaviours we asked respondents about can be found either in the manuscript or in the supporting information, where there is a table of excluded behaviors. Our classification of behaviors as well-established or not, does not preclude future researchers from presenting the behaviors in a  replicated study. It is not our opinion on the behaviours that matters but the consensus achieved on survey analysis. A less well-established behavior may have been just as likely to reach a high validity and consensus as our ‘well-established’ behaviors.

Line 209: "[18] [20]" Should be "[18, 20]". The same error occurs elsewhere in the manuscript.

These formatting errors have been corrected.

Lines 218-220: Given the small number of respondents, I think that the authors might also report the actual number of each category of respondent.

Given that we have tended to use percentages elsewhere in reporting we would prefer to stick with percentages here, rather than amending some and not others. Percentages, we feel gives a better idea of the proportions of each category easily, which a number doesn’t.  

Similarly, I am surprised that the authors did not make any effort to gauge the respondents' understanding of animal welfare, a priori. For example, there is still a view held that welfare is all or principally about physical health that does not include the mental state; this view is slowly being changed, but it would be reassuring if the perspective of the respondents was clarified, particularly as some are self-selecting. The definition of Mormede et al (2018) includes mental, physiological and behavioural needs.

This would have been very interesting to do but probably a little hard to gauge using a quantitative survey format since the questioning could well have influenced respondent's answers. This would be better achieved through use of textual analysis or focus group/interview where the data would be much richer and the findings likely much more robust. We also provided definitions of affective state, and highlighted the 5 domains and EU Welfare Quality Protocols in the preamble to the survey, and this would have also likely biased answers on respondents’ understanding of welfare. It is of course more challenging for people to link a behavior with a valence of affective state, even if they are aware of formal definitions (which is a somewhat academic exercise) since this requires some meta-cognitive analysis. We do see value in referring to this point and have included a note in the discussion about its value in future surveys of this type (lines 433-437).

Table 3: "Interaction with Transparent Boundaries" is this what is referred to earlier as ITB (see Hyperactivity)? If so, include the abbreviation after "Interaction with Transparent Boundaries" to facilitate comprehension.

The abbreviation has been added to table 3.

Line 302: Given the absence of the Lambert et al (2019) review, I am not convinced that this article is the "first step" (see a similar comment re the Conclusions".

The wording ‘first’ has been amended to ‘next’ on all occasions mentioned.

Lines 308-309: Yes, these behaviours are frequently utilised in assessing farm animal welfare; but they are not the only ones. In my opinion, the authors have expressed this thought rather crudely that might lead a reader to understand this text as meaning that these are the only types of indicator used to assess farm livestock welfare, and that is not correct.

We are not clear what this comment relates to since we do not mention farm animals in this section. The phrasing as written is ‘These indicators are of the type that are utilized widely in welfare assessment tools across many domesticated species, and are in many ways focused around health”. We do not think that this statement is incorrect and in our opinion the use of the terminology ‘widely-used’ does not infer exclusivity as suggested. It relates more to the frequency of use.

Line 310: "health". Given my comment at 218-220, the authors have a similar thought to me, and so some clarification of how the respondents defined welfare would have been useful in the contextualisation of their views.

A definition of welfare has been included in the introduction. We believe, like many others, that health is a component of welfare, but that an exclusive focus on health parameters would not adequately measure an animal’s welfare by failing to incorporate those aspects of positive emotional state.

Lines 357-358: But you have just stated that "... handling is likely to be stressful ..." (Line 345), so what do you mean by "human-animal interactions"? Some clarification is required.

Human-animal interactions in zoos are of course many, and do not always involve direct contact in terms of handling, but indirect contact such as visitors’ interactions with animals. The statement here is around generic potential indicators of positive emotions so we do not feel that it needs excessive detail but we have added some clarification of what we mean, with an extra reference.

Lines 361-368: Another possible reason for the discrepancy might also be the understanding of what is meant be welfare on the part of the respondents (as mentioned earlier by the authors, Lines ), specifically relating to health and ignoring the mental or affective elements.

Refer to earlier comments. A definition of welfare has been included in the introduction and further discussion around this issue based on the comments raised has been included in the manuscript.  

Line 364: "posed" I think that the authors probably mean "posited".

 The wording has been amended to ‘posited’.

Lines 376-377: The authors have presented some useful reptile-focused references relating to the applications of these methods. In my view, in the light of limited experimental work on reptiles, it could be useful to include also some key papers of a more fundamental nature so that readers who might wish to pursue these experimental paradigms have key work identified.

We have added an extra paragraph in the introduction (lines 128-148) with some more references from the reptile welfare literature, providing a basis for the derivation of the indicators used in our survey.

 Lines 384-385: Here, the authors are exclusively equating welfare with "a positive or negative emotional state"; this is to ignore elements of physical health that are not linked with emotion (see my earlier comments that refer to the definition proposed by Mormede et al). Are the authors discounting all negative aspects of an animal's state that do not affect its emotion at the moment of assessment? This question returns to my earlier point: how do authors define welfare? and, how do the respondents define welfare? Without that common basis, I think that there are inevitable difficulties of interpretation.

The reviewer is correct and we have perhaps inadvertently limited welfare to emotion here, when clearly our measures do not just reflect emotion. A paragraph on welfare definition has been included in the introduction. Moreover, on checking the survey preamble we couched validity in terms of reliability in assessing animal welfare, without reference to emotion. Therefore, we have deleted the wording ‘i.e. a positive or negative emotional state’ here since it may cause confusion.

The authors point to some of these challenges; but these are not new (Mason & Mendl, 1993) and have been addressed by numerous authors (e.g. Boissy, Mason, Mendl, Paul) in various publications since then.

 We agree that the challenges are not new and yes they are referred to by many authors.  I don’t believe we have suggested that we are the first to identify the issue. Since the discussion here relates to our study and our interpretation on the findings it probably does not need further referencing. 

Mason, G. and Mendl, M. 1993. Why is there no simple way of measuring animal welfare? Animal Welfare 2: 301-319.

Line 461: As I have indicated at several points, I have significant concerns that the respondents did not share a common view of what animal welfare actually is, and so their identification of indicators that might be useful in assessing welfare is put into doubt.

As discussed earlier, the respondents were provided with definitions of affective state and notes on the welfare quality and 5 Domains as part of the survey pre-amble. We would argue that getting respondents to provide a formal definition of welfare is a somewhat academic exercise and does not necessarily influence their ability to link behaviours with a particular affective state. In our experience experienced animal carers are remarkably good at identifying behaviors that are indicative of negative affective states, perhaps linking them to the context in which the behaviour is being performed. By nature of their roles they are also likely to be considering issues of welfare on a daily basis.  Admittedly, identifying behaviours indicative of positive affective states may be relatively more challenging.

Line 464: Poor English: "... behavioral-based indicators almost reached consensus..." Should be: ... behavior-based indicators almost reached consensus..."

This wording has been amended each time it occurs.

 Line 467: As mentioned earlier, other work has already addressed sentience in reptiles, and so I am not convinced that this study represents the "first step".

We appreciate that there have been a number of excellent reviews on sentience in reptiles. We feel that the terms sentience and welfare assessment generate different ideas for people. The need for welfare assessment is predicated on the belief that animals are sentient, but evidence for sentience does not necessarily guide ability to assess welfare in a practical fashion. Nevertheless, the reviewer is correct that ‘first’ is an overstatement and this has been amended to ‘next’step.

 Line 512: "C-well" What is this?

I’m unclear what this comment relates to. This is part of the reference and is the name of the welfare assessment tool the authors derived.

Round 2

Reviewer 2 Report

In my opinion, the authors have addressed my main concerns. I have made a few, minor comments below that I would urge the authors to consider as, in my view, they could improve the MS further.

Line 78: “strength or arousal” should that be “strength of arousal”

Snowball sampling: I am aware of the use of this method of sampling. However, it does have the potential for unconscious bias (e.g. emailing like-minded contacts) and I would still urge the authors in their critique of the study to be a little more reflective.

I think that the paper would read better if there was some clarification of what the authors mean by “less well-established”: a brief clarification would remove a hanging, unanswered question.

Lines 218-220: Given the small number of respondents, I think that the authors might also report the actual number of each category of respondent.

Given that we have tended to use percentages elsewhere in reporting we would prefer to stick with percentages here, rather than amending some and not others. Percentages, we feel gives a better idea of the proportions of each category easily, which a number doesn’t.

My point was not meant to be an either/or, it was to include both the percentage and the actual number because percentages always present challenges to interpretation when small numbers are involved.

Author Response

Line 78: “strength or arousal” should that be “strength of arousal”

I think this is correct as is. We are trying to explain the terminology arousal since readers may not be aware of the meaning in the context of affective states. Strength is a lay term for 'arousal'.

Snowball sampling: I am aware of the use of this method of sampling. However, it does have the potential for unconscious bias (e.g. emailing like-minded contacts) and I would still urge the authors in their critique of the study to be a little more reflective.

A few sentences have been added at lines 490-494 to mention potential issues with snowball sampling. 

I think that the paper would read better if there was some clarification of what the authors mean by “less well-established”: a brief clarification would remove a hanging, unanswered question.

A definition of these has been added at lines 239-241.

Lines 218-220: Given the small number of respondents, I think that the authors might also report the actual number of each category of respondent.

The numbers have been added in addition at lines 272-276.